# ∆-PINNs: PHYSICS-INFORMED NEURAL NETWORKS ON COMPLEX GEOMETRIES

## ABSTRACT

Physics-informed neural networks (PINNs) have demonstrated promise in solving forward and inverse problems involving partial differential equations. Despite recent progress on expanding the class of problems that can be tackled by PINNs, most of existing use-cases involve simple geometric domains. To date, there is no clear way to inform PINNs about the topology of the domain where the problem is being solved. In this work, we propose a novel positional encoding mechanism for PINNs based on the eigenfunctions of the Laplace-Beltrami operator. This technique allows to create an input space for the neural network that represents the geometry of a given object. We approximate the eigenfunctions as well as the operators involved in the partial differential equations with finite elements. We extensively test and compare the proposed methodology against traditional PINNs in complex shapes, such as a coil, a heat sink and a bunny, with different physics, such as the Eikonal equation and heat transfer. We also study the sensitivity of our method to the number of eigenfunctions used, as well as the discretization used for the eigenfunctions and the underlying operators. Our results show excellent agreement with the ground truth data in cases where traditional PINNs fail to produce a meaningful solution. We envision this new technique will expand the effectiveness of PINNs to more realistic applications.

## 1 MOTIVATION

Physics-informed neural networks (PINNs) Raissi et al. (2019) are an exciting new tool for blending data and known physical laws in the form of differential equations. They have been successfully applied to multiple physical domains such as fluid mechanics Raissi et al. (2020), solid mechanics Haghighat et al. (2021), heat transfer Cai et al. (2021) and biomedical engineering Kissas et al. (2020); Ruiz Herrera et al. (2022), to name a few. Even though this technique can be used to solve forward problems, they tend to excel when performing inverse problems. Since their inception, there have been multiple attempts to improve PINNs, in areas such as training strategies Nabian et al. (2021); Wang et al. (2022) and activation functions Jagtap et al. (2020).

Despite recent progress, many works still consider simple geometric domains to solve either forward or inverse problems, hindering the applicability of PINNs to real world problems, where the objects of study may have complicated shapes and topologies. In this area, there have been multiple attempts to introduce complexity to the input domain of the neural networks. One approach is to describe the boundary of the domain with a signed distance function Sukumar & Srivastava (2022); Berg & Nyström (2018); McFall & Mahan (2009). In this way, the boundary conditions can be satisfied exactly instead of relying on a penalty term in the loss function. Another approach is to use domain decomposition to model smaller but simpler domains Jagtap & Karniadakis (2020). Coordinate mappings between a simple domain have also been proposed for convolutional Gao et al. (2021) and fully connected networks Li et al. (2022). Nonetheless, all these works demonstrate examples of 2-dimensional shapes. When using PINNs in 3-dimensional surfaces there have been attempts to ensure that the vector fields that may appear in the partial differential equations remain tangent to the surface by introducing additional terms in the loss function Fang et al. (2021); Sahli Costabal et al. (2020). This approach works well when the surface is relatively simple and smooth, specifically when the Euclidean distance of the embedding 3-D space between two points in the domain is similar to the intrinsic geodesic distance on the manifold. However, in several applications the two distances may sensibly differ, as exemplified in Figure 1.

In this work, we propose to represent the coordinates of the input geometry with a positional encoding based on the eigenfunctions of the Laplace-Beltrami operator of the manifold, or the Laplacian in the case of a bounded open domain in the Euclidean space. In this way, points that are close in the geometry will remain close in the positional encoding space. Next, the input of the neural network will be the value of a finite number of the eigenfunctions evaluated at a point in the domain and the output will remain the physical quantity that we are modeling with PINNs. Positional encoding have shown great success in improving the capabilities of neural networks and are used in transformers Vaswani et al. (2017), neural radiance fields Mildenhall et al. (2020) and PINNs Wang et al. (2021a) to name a few. For our positional encoding, the Laplace-Beltrami eigenfunctions can be approximated numerically for any shape with the finite element method. By approximating the eigenfunctions on a mesh we lose the ability to use automatic differentiation to compute the operators of the partial differential equations within existing library codes. However, automatic differentiation could be preserved whenever the finite element library offers such capability, as many existing libraries do. We show that common operators such as the gradient and the Laplacian can be efficiently computed with finite elements applied to the output of the neural network. We demonstrate the capabilities of the proposed method by testing different geometries, such as a coil, a heat sink and a bunny and different physics, such as the Eikonal equation and heat transfer.

**Laplace-Beltrami eigenfunctions**

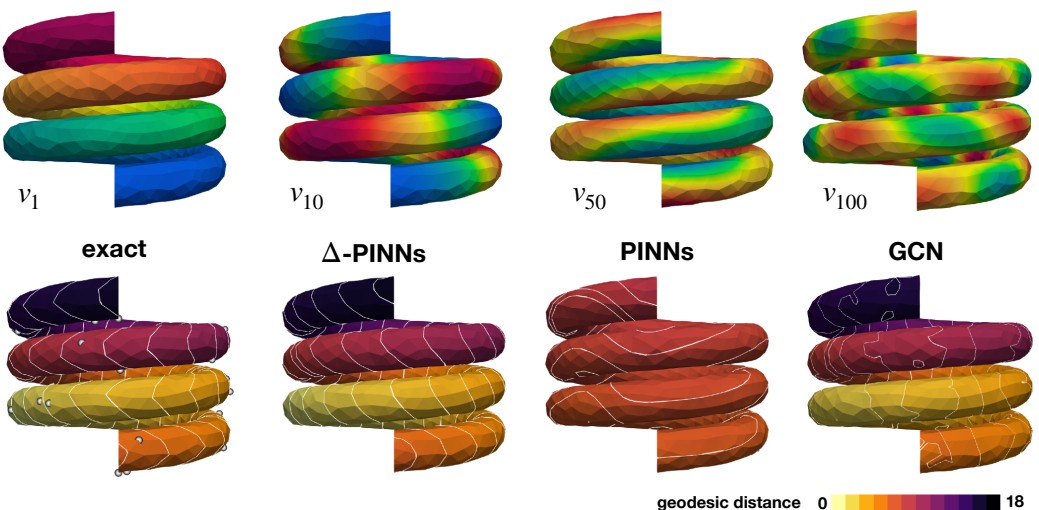

Figure 1: **Learning the Eikonal equation on a coil**. Top row: the 1st, 10th, 50th and 100th Laplace-Beltrami eigenfunction of the geometry. Bottom row, first: the ground truth solution of the Eikonal equation, which represents the geodesic distance, and the data points used for training shown as gray spheres. Second, the solution of Δ-PINNs, our proposed method trained with 50 eigenfunctions, Third, the traditional PINNs approximation. Last, the approximate solution of a physics-informed graph-convolution network.

## 2 METHODS

### 2.1 PHYSICS-INFORMED NEURAL NETWORKS

We consider the problem where we have partial observations of an unknown function $u(\boldsymbol{x})$, with an input domain $\boldsymbol{x} \in \mathcal{B}$, where $\mathcal{B}$ is an open and bounded domain in $\mathbb{R}^d$, $d = 2, 3$ or a $d$-dimensional smooth manifold (typically a surface). We also assume that $u(\boldsymbol{x})$ satisfies a partial differential equation of the form $\mathcal{N}[u, \boldsymbol{\lambda}] = 0$, where $\mathcal{N}[\cdot, \boldsymbol{\lambda}]$ is a potentially non-linear operator parametrized by $\boldsymbol{\lambda}$. The partial observations $u_i$ are located in $\mathcal{B}$ at positions $\boldsymbol{x}_i$, $i = 1, \ldots, N$, which may therefore fall on the boundary $\partial \mathcal{B}$. Boundary conditions such as Neumann boundary conditions $\nabla u \cdot \boldsymbol{n} = g_i$, may also be enforced at boundary points $\boldsymbol{x}_i^b$, $i = 1, \ldots, B$. We proceed by approximating the unknown function with a neural network $u \approx \hat{u} = NN(\boldsymbol{x}, \boldsymbol{\theta})$, parametrized with trainable parameters $\boldsymbol{\theta}$. In order to learn a function that satisfies the observed data, the boundary conditions

and the model, we postulate a loss function composed of three terms

$$\mathcal{L}(\{u_i, \boldsymbol{x}_i\}, \{g_i, \boldsymbol{x}_i^b\}, \boldsymbol{\theta}, \boldsymbol{\lambda}) = MSE_{\text{data}}(\{u_i, \boldsymbol{x}_i\}, \boldsymbol{\theta}) + MSE_{\text{PDE}}(\boldsymbol{\theta}, \boldsymbol{\lambda}) + MSE_{\text{b}}(\{g_i, \boldsymbol{x}_i^b\}, \boldsymbol{\theta}). \tag{1}$$

The first term favours the network to learn the available data

$$MSE_{\text{data}}(\{u_i, \boldsymbol{x}_i\}, \boldsymbol{\theta}) = \frac{1}{N} \sum_i^N (u_i - \hat{u}(\boldsymbol{x}_i; \boldsymbol{\theta}))^2. \tag{2}$$

The second term ensures that the underlying partial differential equation is approximately satisfied

$$MSE_{\text{PDE}}(\boldsymbol{\theta}, \boldsymbol{\lambda}) = \frac{1}{R} \sum_i^R (\mathcal{N}[\hat{u}(\boldsymbol{x}_i^r, \boldsymbol{\theta}; \boldsymbol{\lambda}])^2. \tag{3}$$

Here, we will evaluate the model at locations $\boldsymbol{x}_i^r$ that can be generated randomly within the domain at every optimization iteration or can have predefined values. Minimizing this term will ensure that the model is approximately satisfied in the entire domain. We can also define the parameters of the operator $\mathcal{N}[\cdot, \boldsymbol{\lambda}]$ as trainable variables that will be learned during the optimization procedure. The final term in the loss function ensures that the boundary conditions are satisfied

$$MSE_{\text{b}}(\{g_i, \boldsymbol{x}_i^b\}, \boldsymbol{\theta}) = \frac{1}{B} \sum_i^B (\nabla \hat{u}(\boldsymbol{x}_i^b; \boldsymbol{\theta}) \cdot \boldsymbol{n} - g_i)^2. \tag{4}$$

We note that Dirichlet boundary conditions can be included as data in $\{u_i, \boldsymbol{x}_i\}$. This is the basic structure of physics-informed neural networks that can be accommodated for a large class of problems Raissi et al. (2019). One issue of this formulation is that the neural network that represents the solution is parametrized with a vector $\boldsymbol{x} \in \mathbb{R}^3$, which is a Euclidean space. In this way, it is difficult to inform the network about the geometry of the domain or, in the case of a complex domain in $\mathbb{R}^d$ where the boundaries lie. This is important in the computation of the differential operators. Furthermore, points in the manifold $\mathcal{B}$ that are close in the Euclidean space are not necessarily close in the intrinsic distance, as illustrated in Figure 1.

## 2.2 Eigenfunctions of Laplace-Beltrami operator

In this work, we propose to encode the manifold of interest $\mathcal{B}$ with the eigenfunctions of the Laplace-Beltrami operator $-\Delta[\cdot] = \nabla \cdot \nabla[\cdot]$, instead of using the Euclidean coordinates $\boldsymbol{x}$. These functions $v_i(\boldsymbol{x})$ can be computed by solving the eigenvalue problem: $-\Delta v_i = \lambda_i v_i$, where $\lambda_i$ is the correspondent eigenvalue of $v_i(\boldsymbol{x})$. In the case of a domain with open boundaries, this problem is usually accompanied with either homogeneous Dirichlet or Neumann boundary conditions. In this work, we will use the latter. The set of eigenvalues provide a spectral signature of the domain, albeit not unique Gordon et al. (1992). It is possible to show that the eigenvalues of the Laplace-Beltrami operator, endowed with homogeneous boundary conditions, are real, non-negative, and tending to infinity. When ordered by their magnitude, the first eigenvalue $\lambda_0$ is always zero, whereas the second one $\lambda_1$ is strictly positive. We note that the physical interpretation that small eigenvalues are associated with low frequency functions, represented by the corresponding eigenfunction. The set of eigenfunctions form an orthogonal basis for the Hilbert space $L^2(\mathcal{B})$, hence every function in $L^2(\mathcal{B})$ can be expressed as a linear combination of the eigenfunctions Evans (2010). Although these functions may be computed analytically for a few shapes (e.g., equilateral triangle, rectangle, circle), their numerical approximation is possible for an arbitrary manifold. Here, we represent the domain $\mathcal{B}$ using a mesh with nodes and elements and solve the eigenvalue problem using finite element shape functions. As such, we can obtain the discrete Laplacian matrix $\boldsymbol{A}$ and mass matrix $\boldsymbol{M}$ as

$$\boldsymbol{A}_{ij} = \mathop{\mathbf{A}}_{\text{e}=1}^{n_{\text{el}}} \int_{\mathcal{B}} \nabla N_i \cdot \nabla N_j d\mathcal{B}, \quad \boldsymbol{M}_{ij} = \mathop{\mathbf{A}}_{\text{e}=1}^{n_{\text{el}}} \int_{\mathcal{B}} N_i N_j d\mathcal{B}, \tag{5}$$

where $\mathbf{A}$ represents the assembly of the local element matrices, and $N$ are the finite element shape functions. Then, we solve the eigenvalue problem: $\boldsymbol{A}\boldsymbol{v} = \lambda \boldsymbol{M}\boldsymbol{v}$, see Appendix A for more details.

## 2.3 PHYSICS-INFORMED NEURAL NETWORKS ON MANIFOLDS

We propose to represent the position of a point in the manifold by $N$ eigenfunctions of the Laplace-Beltrami operator associated with the $N$ lowest eigenfunctions, such that $\boldsymbol{v}_i = [v_1(\boldsymbol{x}_i), v_2(\boldsymbol{x}_i), ..., v_N(\boldsymbol{x}_i)]$. Now, instead of parametrizing the neural network with the Euclidean position, we use: $u \approx \hat{u} = NN(\boldsymbol{v}(\boldsymbol{x}), \boldsymbol{\theta})$. Since $\boldsymbol{v}$ varies smoothly in the manifold, intrinsically close points will be also close in the space $\boldsymbol{v}$.

Another interpretation of the proposed method is that we are constructing features or positional encodings for points in the manifold. Then, the approximation of the solution $\hat{u}$ can be seen as a non-linear combination of these features. In our case, the eigenfunctions represent features of increasing frequency, as seen in Figure 1. A similar concept has been proposed for Cartesian domains with Fourier feature mappings Tancik et al. (2020); Wang et al. (2021a). Here, the Cartesian coordinates are transformed with sine and cosine functions, which have random frequencies that can be tuned for the problem of interest. The main difference is that in our method the frequencies are given by the eigenfunctions, and are defined in the manifold where want to solve the problem. We can select different eigenfunctions depending on the nature of problem, where high frequency problems might need a larger amount of eigenfunctions.

The main challenge of our methodology is that the differential operators cannot be computed directly from the neural network via automatic differentiation. Since the eigenfunctions cannot be computed in closed form for an arbitrary manifold, we have a map between $\boldsymbol{x}$ and $\boldsymbol{v}$ that is defined using finite elements. The only way to differentiate the eigenfunctions $\boldsymbol{v}$ with respect to the position $\boldsymbol{x}$ is again by using finite elements. Therefore, we use this numerical approximation of the differential operators that are required to compute the PDE operator $\mathcal{N}[\cdot]$. Commonly used operators, such as the gradient and the Laplacian can be easily computed with linear finite elements. We also note that one could employ the chain rule to compute, for example, the gradient, such that $\nabla_{\boldsymbol{x}} \hat{u} = \nabla_{\boldsymbol{v}} \hat{u} \cdot \nabla_{\boldsymbol{x}} \boldsymbol{v}$. The term $\nabla_{\boldsymbol{v}} \hat{u}$ can be computed with automatic differentiation since it only depends on the neural network, and the second term $\nabla_{\boldsymbol{x}} \boldsymbol{v}$ needs to be computed with finite elements. We found small advantages with this approach, and it becomes cumbersome for second order operators. Instead, we take the simplest route: we use the neural network predictions at the nodal locations and use finite elements to evaluate the operator. For instance, to compute $\nabla_{\boldsymbol{x}} \hat{u}$ at the centroid of a linear triangular element, we need to evaluate the neural network $\hat{u}_i$ at every $\boldsymbol{v}_i$, which corresponds to the selected eigenfunctions evaluated at the nodal locations $\boldsymbol{x}_i$. Then, the gradient can be computed as $\nabla_{\boldsymbol{x}} \hat{u} \approx \boldsymbol{B}^e \cdot [\hat{u}_1, \hat{u}_2, \hat{u}_3]^T$, where the matrix $\boldsymbol{B}^e$ defines the gradient operator for a linear triangular element in particular, see Appendix A. One of the advantages of this approach is that the gradient is guaranteed to be tangent to the manifold, which is not the case for the traditional physics-informed neural network formulation Sahli Costabal et al. (2020). Similarly, we can compute the Laplacian $\Delta \hat{u}_i$ at any given nodal location by predicting $\hat{u}_j$ at all the neighboring nodes, including the node of interest. Then, we can use the definition of the discrete Laplacian presented in equation 5 and approximate $\Delta \hat{u}_i \approx \boldsymbol{A}_{ij} \hat{u}_j$.

The advantage of being able to compute the operators locally at the element or node level is that we still use mini-batch techniques to estimate the loss function and its gradient. In this way, we avoid predicting the operator for the entire manifold, which could slow the training process for large meshes. In general, we will randomly select nodes or elements for each training iteration to evaluate the loss term $MSE_{\text{PDE}}$ defined in equation 3.

## 3 NUMERICAL EXPERIMENTS

### 3.1 EIKONAL EQUATION ON A COIL

In our first numerical experiment we compute geodesic distances using the Eikonal equation

$$\sqrt{\nabla u \cdot \nabla u} = 1, \ u(\boldsymbol{x}_b) = 0, \tag{6}$$

This equation has multiple applications, for instance, in cardiac electrophysiology Sahli Costabal et al. (2020) and seismology Smith et al. (2020), because it can be used to model the arrival times of a traveling wave. In the particular form shown in equation 6, the solution $u(\boldsymbol{x})$ can be interpreted as the geodesic distance from the point $\boldsymbol{x}_b$ to any point $\boldsymbol{x}$ on the manifold.

In this example, we will solve this equation on the surface of a coil, which is generated by an helix with 30 mm of diameter and 12 mm per revolution of pitch. This curve is extruded with a circular cross-section with 5 mm of radius. The resulting geometry can be seen in Figure 1, which is discretized with 1546 points and 3044 triangles. We can generate a ground truth solution of the Eikonal equation by randomly selecting a point on the mesh for $x_b$ and then applying the exact geodesic algorithm Mitchell et al. (1987) as implemented in libigl Jacobson et al. (2018). We randomly select 40 points of this solution to use as data $u_i$. In this case, the partial differential equation loss defined in 3 takes the form

$$MSE_{\text{PDE}}(\boldsymbol{\theta}) = \frac{1}{R} \sum_i^R \left( \sqrt{(\boldsymbol{B}_r^e \hat{\boldsymbol{u}}_r^e) \cdot (\boldsymbol{B}_r^e \hat{\boldsymbol{u}}_r^e)} - 1 \right)^2, \tag{7}$$

where $\boldsymbol{B}_r^e \hat{\boldsymbol{u}}_r^e$ is the approximation of the gradient for a particular triangle $r$. Here, the vector $\hat{\boldsymbol{u}}_r^e$ is the prediction of the neural network at the nodes of the triangle $r$, which depend on the trainable parameters $\boldsymbol{\theta}$. We use 50 eigenfuctions of the Laplace-Beltrami operator as input of the neural network, which is then followed by a single hidden layer with 100 neurons and hyperbolic tangent activations. We train the neural network with the ADAM optimizer Kingma & Ba (2014) for 40,000 iterations with default parameters. We use a batch size of 10 samples to evaluate both the $MSE_{\text{data}}$ and $MSE_{\text{PDE}}$ term. For comparison, we also train the same neural network by removing the physics, that is not considering the partial differential equation loss shown equation 7. Additionally, we train a regular physics-informed neural network, where we use the Cartesian coordinates as the input of the neural network. We implemented the Eikonal equation in the $MSE_{\text{PDE}}$ computing $\nabla u$ through automatic differentiation. This network has 2 hidden layers with 50 and 100 neurons, respectively. We also tested against a physics-informed graph neural network Gao et al. (2022), detailed in Appendix B.1, with matched number of parameters. We train all methods for the same number of iterations and learning rate.

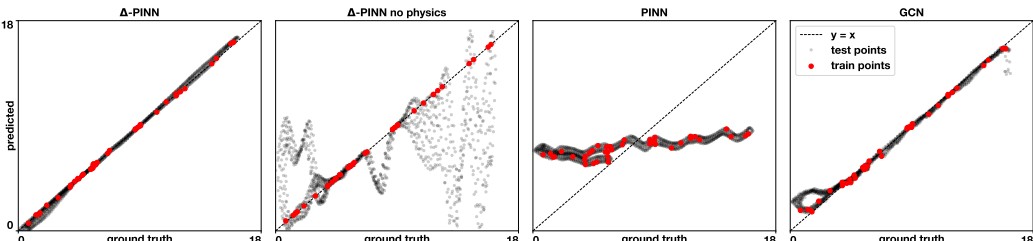

Figure 2: **Accuracy the coil example with the Eikonal equation.** Correlation between predicted and ground truth values of geodesic distance for $\Delta$-PINNs (first), $\Delta$-PINNs trained without the loss functions that includes the Eikonal equation (second), traditional PINNs (third) and a graph convolutional network (last).

Figures 1 and 2 show the results of these experiments. For $\Delta$-PINNs and the graph neural network we select the best out of 5 random initializations (see Appendix B.1). In the bottom row of Figure 1, we see the exact solution to the geodesic problem, the physics-informed neural network learned solution, the graph convolutional network and the proposed method. On one hand, we see that PINNs fail to capture the fast variations in the Cartesian space and then to predict a nearly constant intermediate value. On the other hand, we see that $\Delta$-PINNs can represent the exact solution with high accuracy. The graph convolutional network performs better than PINNs but struggle to correctly learn the level-sets of the solution. This is reflected in Figure 2, where we show the correlation between the predicted and ground truth data for train and test points. We see that $\Delta$-PINN achieves a high correlation in both training and testing data, while PINNs fail to fit both. When the physics are removed from $\Delta$-PINNs, the method can only overfit the training data with a poor performance in the test data. Qualitatively, the normalized mean squared errors on the combined train and test dataset correspond to $3.68 \cdot 10^{-3}$ for $\Delta$-PINNs, 0.99 for for $\Delta$-PINNs without physics, 1.01 for PINNs, and $2.95 \cdot 10^{-2}$. These results show more than 2 orders of magnitude improvement for the proposed method over the traditional formulation, at the expense of small pre-processing step of computing the eigenfunctions of the Laplace-Beltrami operator for that geometry (see Appendix E). It is worth noting that generating the solution of the Eikonal equation with the exact geodesic algorithm does not lead to a zero $MSE_{\text{PDE}}$ loss as defined in equation 7 due to the underlying numerical

discretization. If we evaluate this term using all the points in the mesh we obtain a residual value of $2.11 \cdot 10^{-2}$. Hence, the proposed method works even in the presence of imperfect physics, when the model is not exactly satisfied.

## 3.2 HEAT TRANSFER IN A HEAT SINK

In this experiment, we simulate a heat sink with convection losses in 2 dimensions and steady state. This can be expressed as the following boundary value problem

$$\begin{aligned} \Delta u(\boldsymbol{x}) &= 0, \ \boldsymbol{x} \in \mathcal{B}; \quad u(\boldsymbol{x}) = 1, \ \boldsymbol{x} \in d\mathcal{B}_D, \\ \nabla u(\boldsymbol{x}) \cdot \boldsymbol{n} &= 0.1u(\boldsymbol{x}), \ \boldsymbol{x} \in d\mathcal{B}_C; \quad \nabla u(\boldsymbol{x}) \cdot \boldsymbol{n} = 0, \ \boldsymbol{x} \in d\mathcal{B}_N. \end{aligned} \tag{8}$$

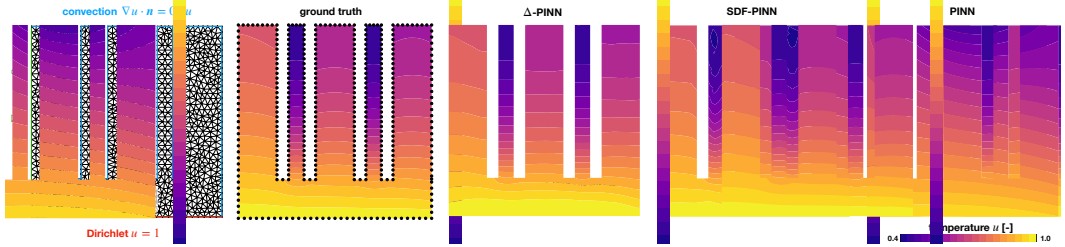

Figure 3: **Learning the temperature distribution of a heat sink from boundary measurements.** First, the boundary conditions and the finite element mesh used to create the solution. Second, the ground truth, and in the learned solution of (third) $\Delta$-PINNs, (forth) signed distance function PINNs and (last) traditional PINNs. The data points are shown on the ground truth panel in black.

We include three different types of boundary conditions: Dirichlet in $d\mathcal{B}_D$, Robin in $d\mathcal{B}_C$ and Neumann $d\mathcal{B}_N$. The Dirichlet boundary condition is to represent another body with constant, higher than ambient, temperature, which the heat sink $\mathcal{B}$ is attempting to dissipate. The Robin boundary condition represents a convective flux, which is proportional to the temperature $u$. The Neumann boundary condition is used to represent the symmetry of the domain. The geometry and location of the boundaries are shown in Figure 3. This example could be interpreted as a simplified model for the heat sink that is typically placed above a central processing unit (CPU) of a computer. To generate a ground truth solution for this problem we performed finite element simulations using the FEniCS library Logg et al. (2012) on a mesh with 1289 nodes and 2183 linear triangular elements. In this experiment we will attempt to solve the inverse problem of learning the temperature inside the domain $\mathcal{B}$ from the information of the temperature in the boundary $d\mathcal{B}$. We will again use the proposed $\Delta$-PINNs with and without physics, as well conventional PINNs to solve this problem. In all cases, we will not inform the type of boundary conditions applied, and instead we only provide temperature measurements $u_i$ at the boundary $d\mathcal{B}$. Thus, we will not consider the term $MSE_\mathrm{b}$ in the loss function defined in equation 1. In practice, this is important because the true form of the convective fluxes is often unknown or empirically approximated. For $\Delta$-PINNs we will approximate the Laplacian operator $\Delta u$ with triangular finite elements using the definition of discrete Laplacian $\boldsymbol{A}$ shown in equation 5. Then, we can define the loss term

$$MSE_\mathrm{PDE}(\boldsymbol{\theta}) = \sum_i^R \left( \sum_{j \in N(i)} A_{ij} \hat{u}_j \right)^2, \tag{9}$$

where the index $i$ represents the nodes of the mesh inside the domain $\mathcal{B}$, $j \in N(i)$ represents the neighbor nodes of the node $i$, including itself. The term $\hat{u}_j$ represents the output of the neural network at the nodal locations $j$ that depends on trainable parameters $\boldsymbol{\theta}$. For traditional PINNs, we use automatic differentiation to compute $\Delta\hat{u}$ and construct the loss term $MSE_\mathrm{PDE}$. We also compare to physics-informed neural network that exactly imposes the boundary data using signed distance functions Sukumar & Srivastava (2022), which can only work in the case of a flat surface or 3D solid, detailed in Appendix B.2. We use 50 eigenfunctions to represent the domain $\mathcal{B}$ for $\Delta$-PINNs, and 3 hidden layers of 100 neurons for all cases. We train the network with ADAM Kingma & Ba (2014) for 50,000 iterations using a batch size of 30 samples to evaluate each of the terms of the loss

function. We use 393 values of $u$ located at the boundaries as training data, as shown in Figure 3. We note that we use the same operator, the Laplacian, for the PDE that governs this problem and to construct the input of the neural network. However, the eigenfunctions are computed with homogeneous Neumann boundary conditions, which guarantees that the eigenfunctions $v_i$ are not solutions to the boundary value problem shown in 9 since they have different boundary conditions.

The results of this experiment are summarized in Figures 3 and 4. First, we see that the geometry of the heat sink greatly influences the solution, as shown in Figure 3. The narrow fins have more temperature reduction than the wide ones. Qualitatively, we observe that $\Delta$-PINN produces the closest temperature distribution to the ground truth solution. The method is able to produce different temperature distributions for the different fins, which may be close in Euclidean distance, but are far apart in the intrinsic distance. The signed distance function PINNs produce a reasonable approximation everywhere except for the narrow fins, where there is an artificial curvature. Finally, traditional PINNs tend to favor satisfying the $MSE_{\text{PDE}}$, producing effectively a nearly linear solution for which trivially $\Delta\hat{u} = 0$. However, in this process the data provided at the boundaries is not fitted. Also, this method is not able to produce different temperature profiles for the different fins, correlating points that are close only in the cartesian space. Quantitatively, we see that the normalized mean squared error on the combined train and test sets drops 2 orders of magnitude for $\Delta$-PINNs ($1.62 \cdot 10^{-3}$) when compared to traditional PINNs ($1.48 \cdot 10^{-1}$). The signed distance function PINN produces an error slightly lower than $\Delta$-PINNs of $1.45 \cdot 10^{-3}$, through the exact imposition of the data as boundary conditions. Also in Figure 4, we can see that for PINNs, the correlations reflect exactly structure of the different fins. With this example, we have shown a problem with simple physics, a linear inverse problem, with some degree of complexity in the geometry of the domain may cause PINNs to fail while $\Delta$-PINN may successfully work.

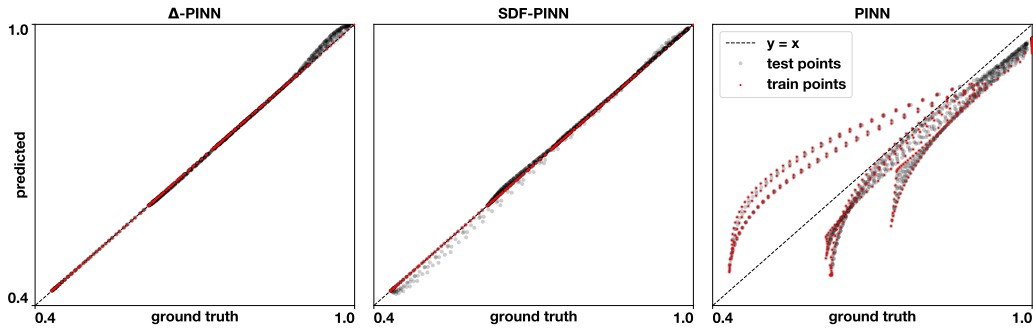

Figure 4: **Accuracy the heat sink example.** Correlation between predicted and ground truth values of temperature for $\Delta$-PINNs (left), signed distance function PINNs (center) and traditional PINNs (right).

### 3.3 PHYSICS-INFORMED DEEP OPERATOR NETWORK

We finalize this section by presenting a deep operator network Lu et al. (2019); Wang et al. (2021b) that learns the geodesic distance between two points on a manifold. This type of networks learn to transform functions instead of single points. They are composed by a branch network that encodes the input functions and a trunk net that encodes the input position. The input function in our case corresponds to boundary condition of the Eikonal equation, shown in equation 6 and the input coordinate corresponds to points in the manifold. We note two properties of the geodesic distance function $d(\cdot, \cdot)$: first, it is symmetric, $d(\boldsymbol{x}_1, \boldsymbol{x}_2) = d(\boldsymbol{x}_2, \boldsymbol{x}_1)$, since the distance from $\boldsymbol{x}_1$ to $\boldsymbol{x}_2$ is the same as the distance from $\boldsymbol{x}_2$ to $\boldsymbol{x}_1$. Second, the distance from a point to itself is zero, that is $d(\boldsymbol{x}, \boldsymbol{x}) = 0$ for every point in the manifold. As in the rest of the experiments, we will represent points in the manifold by the eigenfunctions of the Laplace-Beltrami operator $\boldsymbol{v}_i = [v_1(\boldsymbol{x}_i), v_2(\boldsymbol{x}_i), ..., v_N(\boldsymbol{x}_i)]$. With this is mind, we propose the following deep operator network architecture $\hat{d}$ to approximate $d$

$$\bar{d}(\boldsymbol{v}_1, \boldsymbol{v}_2) = NN(\boldsymbol{v}_1; \boldsymbol{\theta}_T) \cdot NN(\boldsymbol{v}_2; \boldsymbol{\theta}_B), \tag{10}$$

$$\hat{d}(\boldsymbol{v}_1, \boldsymbol{v}_2) = \frac{1}{2} \left(1 - \frac{\boldsymbol{v}_1 \cdot \boldsymbol{v}_2}{\|\boldsymbol{v}_1\| \|\boldsymbol{v}_2\|}\right) (\bar{d}(\boldsymbol{v}_1, \boldsymbol{v}_2) + \bar{d}(\boldsymbol{v}_2, \boldsymbol{v}_1)), \tag{11}$$

where $\boldsymbol{\theta}_T$ and $\boldsymbol{\theta}_B$ represent the trainable parameters of the trunk and branch networks respectively. This architecture automatically satisfies the symmetry and zero distance properties of the geodesic distance function. The network will be informed by informed by the Eikonal equation, which should be satisfied at any point of the domain. In this case, we use the chain rule to compute the gradient $\nabla_{\boldsymbol{x}_1}\hat{d} = \nabla_{\boldsymbol{v}_1}\hat{d} \cdot \nabla_{\boldsymbol{x}_1}\boldsymbol{v}$. The gradient of the eigenfuctions with respect to the input positions $\nabla_{\boldsymbol{x}_1}\boldsymbol{v}$ is computed with finite elements and gradient of the predicted distance function with respect to the input eigenfunctions $\nabla_{\boldsymbol{v}_1}\hat{d}$ is computed with automatic differentiation. The loss function to train the neural network still consists of the one detailed in equation 1, but in this case, we do not have boundary data and the partial differential equation loss takes the form:

$$MSE_{\mathrm{PDE}}(\boldsymbol{\theta}_T, \boldsymbol{\theta}_B) = \frac{1}{R}\sum_i^R \left( \sqrt{(\nabla_{\boldsymbol{x}_1}\hat{d} \cdot \nabla_{\boldsymbol{x}_1}\hat{d})} - 1 \right)^2, \tag{12}$$

which is evaluated at the $R$ triangle centroids of the mesh using a mini-batch strategy.

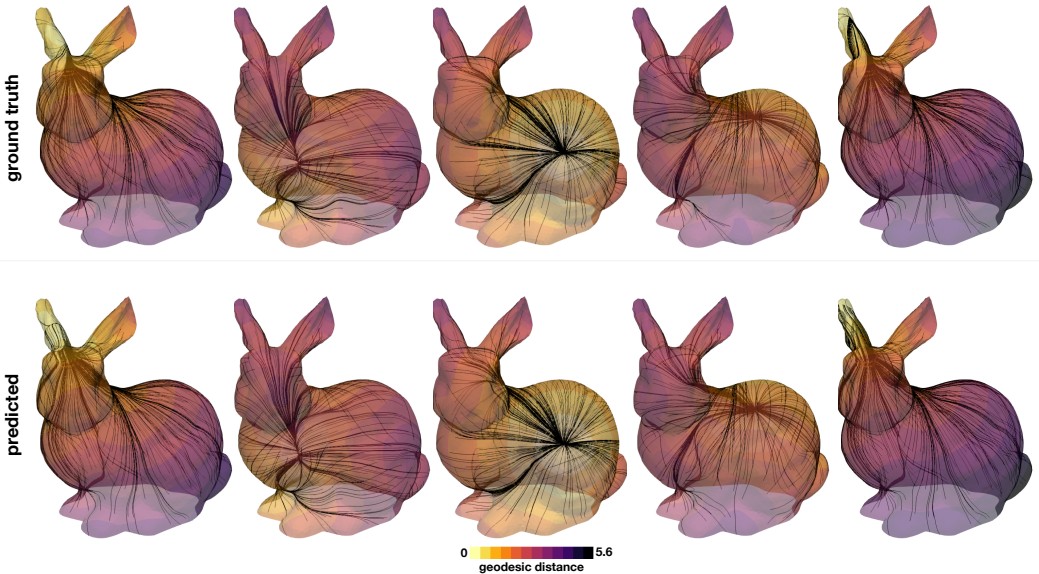

Figure 5: **Geodesics on a bunny.** We learn the geodesic distance between any two points in the bunny surface. We randomly select 5 points and show the ground truth and predicted geodesic distance between these points and the rest of the surface, and geodesic paths between 300 randomly selected points and these origin points.

We test the proposed methodology in the Stanford bunny Turk & Levoy (1994), obtained from the software MeshMixer. This mesh has 5,550 vertices and 10,996 triangles, which results in 15,404,025 pairs of distance to be learned. We randomly select 50,000 pairs (0.32% of the total pairs) as training data, for which we compute the exact geodesic distance Mitchell et al. (1987). We use the 50 eigenfunctions associated with the lowest eigenvalues of the Laplace-Beltrami operator to define $\boldsymbol{v}(\boldsymbol{x}_i)$. Both for the trunk and branch network, we use 10 hidden layers of 200 neurons each. We train for 50,000 iterations with ADAM Kingma & Ba (2014) with exponential decay of the learning rate. To demonstrate the potential of this method, we also compute geodesic paths between $\boldsymbol{x}_1$ and $\boldsymbol{x}_2$ by solving the following initial value problem: $\dot{\boldsymbol{x}} = -\nabla_{\boldsymbol{x}}d(\boldsymbol{x}, \boldsymbol{x}_2)$, $\boldsymbol{x}(0) = \boldsymbol{x}_1$. Some representative results of this method are shown in Figure 5, where we display the geodesic distance from 5 randomly selected points to the rest of the manifold for our method and the ground truth. We also show geodesic paths between 300 randomly selected points to each of the 5 origin points. We see qualitatively very good agreement in both for the predicted geodesic distances and geodesic paths. Furthermore, the geodesic path are actually directed to the origin point, which shows that the learned geodesic distances continuously grow out of the origin point. Quantitatively, we evaluate the test error using 10,000 randomly selected pairs and obtained a mean absolute error normalized by the maximum geodesic distance of 0.57%.

## 4 DISCUSSION

In this work, we present a novel method to use physics-informed neural networks in complex geometries by representing manifolds by the eigenfunctions of the Laplace-Beltrami operator. Given a mesh representing the geometry, we can compute these functions with standard finite elements. We show that this method outperforms the original formulation of PINNs, especially in cases where there points in the input domain are close in Euclidean distance but far in the intrinsic distance. We test our methodology by solving an inverse problem with the Eikonal equation and the heat equation, a forward Poisson problem, and an operator learning problem for inferring the geodesic distance between two points on a complex manifold. We show better performance than a physics-informed graph convolutional network Gao et al. (2022) and comparable performance to PINNs with signed distance functions, which is specially tailored for 2D problems with boundary data Sukumar & Srivastava (2022). The Laplace-Beltrami eigenfunctions have been used in other applications of machine learning, such as point signatures in shapes Rustamov et al. (2007); Sun et al. (2009), shape descriptors Fang et al. (2015), for neural fields Koestler et al. (2022) and to approximate kernels on manifolds for Gaussian process regression Borovitskiy et al. (2020) and classification Gander et al. (2022). To our knowledge this is the first time they have been used in the context of physics-informed neural networks.

The proposed method can be seen as an extension of positional encoding methods Vaswani et al. (2017), such as Fourier features Tancik et al. (2020), to arbitrary geometries. Furthermore, combining the Laplacian eigenfunctions in $\mathbb{R}^N$, we can recover a series of cosine functions when using Neumann boundary conditions and sine functions when using Dirichlet boundary conditions, which correspond to the encodings proposed in Vaswani et al. (2017). These techniques have significantly improved the performance of neural networks, including PINNs Wang et al. (2021a). Nonetheless, we observed in our experiments that only using the eigenfunctions from the Laplacian with Neumann boundary conditions was enough to obtain accurate results. We see that including the physics of the problem greatly improves the results. Furthermore, we tested our method in a pure regression tasks in D and compared it to Gaussian process regression which can be naturally extended to work on manifolds. We see that neural networks with the eigenfunctions as input tend to overfit the data and the results depend heavily on the number of eigenfunctions used. This confirms the regularization effect of the physics, explaining the excellent performance in problems shown here. We also note that Gaussian process are an effective tool for regression on manifolds Borovitskiy et al. (2020), but extending them to include physical knowledge expressed as non-linear constraints is cumbersome, as it requires approximate inference techniques and will inevitably suffer from the classical poor scaling of Gaussian processes Raissi et al. (2017).

In our experiment for the Poisson equation where we can directly compare to traditional PINNs (see Appendix C), we observe that is possible to obtain similar levels of errors, showing that our method can also be used for simple domains. Furthermore, we did not see a decrease in performance when we compared the numerically obtained eigenfunctions versus the exact ones. The number of eigenfunctions used did alter the accuracy of our method. This quantity is a hyper-parameter that needs to be tuned, as for all common positional encodings Tancik et al. (2020). We also observed that the size of the discretization to compute the operators also has an impact on the accuracy in a non-monotonic fashion. The way that changing the discretization affects the loss landscape and the training dynamics requires further studies which are beyond the scope of this work.

Our current approach has some limitations. We numerically approximate the operators involved in the partial differential equations using finite elements. Although the gradient and the Laplacian, which are the most commonly used operators, can be approximated with linear elements, more sophisticated elements might be needed for other operators. We also need to compute the eigenfunctions of the Laplace-Beltrami operator as a pre-processing step. Nonetheless, as we show in Appendix E, this task takes just a few seconds even for complex meshes with more than $10^5$ nodes.

As future work, we plan to expand this methodology to work on 3D solid geometries, for which we can compute the eigenfunctions of the Laplace operator. There are multiple opportunities in this area, as most partial differential equations are actually solved in this type of domains. We would also like to extend this approach to time dependent problems, where we can include time as an additional feature of the neural network. We envision that $\Delta$-PINNs will enlarge the possibilities for applications of physics-informed neural networks in more complex and realistic geometries.

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

## A   THE USE OF FINITE ELEMENTS IN $\Delta$-PINNS

In this section we provide a brief description of the finite element method in the context of the proposed methodology. In finite elements, we approximate the discretization of a function $u(\boldsymbol{x})$ on a mesh with nodes and elements as the sum of functions with compact support Hughes (2012)

$$u(\boldsymbol{x}) = \sum_i^{n_{\text{nodes}}} N_i(\boldsymbol{x})u_i, \tag{13}$$

where $N_i$ represent the so-called shape functions that are associated with node $i$ of the mesh, and $u_i$ is the value of the function at that node. In this work we only consider Lagrangian shape functions, that is those that satisfy the following delta property

$$N_i(\boldsymbol{x}_j) = \begin{cases} 1, & i = j, \\ 0, & i \neq j. \end{cases} \tag{14}$$

These functions also satisfy the partition of unity, such that $\sum_i^{n_{\text{nodes}}} N_i(\boldsymbol{x}) = 1 \ \forall \boldsymbol{x}$. An example of the shapes functions in 1D would be the hat (also called triangular) function defining a linear finite element. For a surface in 3D, the linear case corresponds to a piece-wise linear function that is 1 in node $i$ and 0 in all neighboring nodes. In our case, we would like to compute the eigenfunctions Laplace-Beltrami operator using finite elements, which corresponds to solving the following boundary value problem

$$\begin{aligned} -\Delta u(\boldsymbol{x}) &= \lambda u(\boldsymbol{x}), \ \boldsymbol{x} \in \mathcal{B}, \tag{15} \\ \nabla u \cdot \boldsymbol{n} &= 0, , \boldsymbol{x} \in d\mathcal{B}. \tag{16} \end{aligned}$$

We can rewrite this equation by multiplying by a test function $w(\boldsymbol{x})$ and integrating in the entire domain

$$-\int_{\mathcal{B}} w(\Delta u + \lambda u)d\mathcal{B} = 0. \tag{17}$$

Using Green's first identity, we arrive at the variational formulation of the problem

$$\int_{\mathcal{B}} (\nabla w \cdot \nabla u - \lambda w u)d\mathcal{B} - \int_{d\mathcal{B}} w \nabla u \cdot \boldsymbol{n} d\mathcal{B} = 0, \tag{18}$$

for all possible choices of the test function $v$ in some function space. The second term vanishes due to the boundary conditions of the problem. Typical choice for the space is the Sobolev space $H^1(\mathcal{B})$ for Neumann boundary conditions and $H_0^1(\mathcal{B})$ for Dirichlet conditions. The space $H^1(\mathcal{B})$ contains the square-integrable functions (in the sense of Lebesgue) with square-integrable gradient.

The Galerkin approach for the numerical approximation of the eigenvalue problem consists in selecting a finite dimensional subspace of $H^1(\Omega)$. Here, we consider the classic finite element space of piecewise linear functions. Now, we approximate both functions with finite elements $w(\boldsymbol{x}) \approx \sum_i^{n_{\text{nodes}}} N_i(\boldsymbol{x})w_i$, $u(\boldsymbol{x}) \approx \sum_j^{n_{\text{nodes}}} N_j(\boldsymbol{x})u_j$. Replacing, we obtain

$$\int_{\mathcal{B}} \sum_i^{n_{\text{nodes}}} \nabla N_i(\boldsymbol{x})w_i \cdot \sum_j^{n_{\text{nodes}}} \nabla N_j(\boldsymbol{x})u_j - \lambda \int_{\mathcal{B}} \sum_i^{n_{\text{nodes}}} N_i(\boldsymbol{x})w_i \sum_j^{n_{\text{nodes}}} N_j(\boldsymbol{x})u_j d\mathcal{B} = 0, \tag{19}$$

and since the nodal values $u_j$, $w_i$ do not depend on $\boldsymbol{x}$ we can rewrite

$$\sum_i^{n_{\text{nodes}}} w_i \sum_j^{n_{\text{nodes}}} u_j \left( \int_{\mathcal{B}} \nabla N_i(\boldsymbol{x}) \cdot \nabla N_j(\boldsymbol{x}) - \lambda \int_{\mathcal{B}} N_i(\boldsymbol{x})N_j(\boldsymbol{x})d\mathcal{B} \right) = 0. \tag{20}$$

Since this equation must hold for any test function, thus any values of $w_i$, we arrive at a system of equations

$$\sum_j^{n_{\text{nodes}}} u_j \left( \int_{\mathcal{B}} \nabla N_i(\boldsymbol{x}) \cdot \nabla N_j(\boldsymbol{x}) - \lambda \int_{\mathcal{B}} N_i(\boldsymbol{x})N_j(\boldsymbol{x})d\mathcal{B} \right) = \boldsymbol{0}. \tag{21}$$

This system can be written in matrix form by defining

$$\boldsymbol{A}_{ij} = \mathop{\mathbf{A}}_{e=1}^{n_{\text{el}}} \int_{\mathcal{B}} \nabla N_i \cdot \nabla N_j d\mathcal{B}, \quad \boldsymbol{M}_{ij} = \mathop{\mathbf{A}}_{e=1}^{n_{\text{el}}} \int_{\mathcal{B}} N_i N_j d\mathcal{B}, \tag{22}$$

where **A** represents the assembly of the local element matrices. Then, we solve the eigenvalue problem: $\boldsymbol{Au} = \lambda \boldsymbol{Mu}$ to obtain the eigenfunctions, where $\boldsymbol{u} = [u_1, ..., u_i, ..., u_{n_{\text{nodes}}}]$.

It is possible to show that the discrete set of eigenfunctions, obtained by solving the problem $\boldsymbol{Au} = \lambda \boldsymbol{Mu}$, converges to the analytical eigenfunctions as the mesh size goes to zero (under some uniformity assumptions in the way the limit is attained.) For linear basis functions and convex domains, the eigenvalues converge quadratically with respect to the mesh size Babuška & Osborn (1991). The convergence of the eigenfunctions is linear in the energy norm and quadratic in the $L^2$ norm. For non-convex domains, the convergence rate might be limited by the regularity of the eigenfunctions (e.g., the L-shaped domain.) Furthermore, the geometrical error in approximating the domain or the surface with a piece-wise linear complex may introduce additional approximation errors, usually in regions where the curvature is large. In this context, isoparametric or isogeometric finite elements are more indicated Buffa et al. (2010).

In this work, we use linear triangular elements, for which we will provide further detail. A important part of the proposed methodology is the computation of the gradient $\nabla u$. In linear finite elements, the gradient is constant inside the element. We compute the gradient in the element as $(\nabla u)^e = \boldsymbol{B}^e \boldsymbol{u}^e$, where

$$\boldsymbol{B}^e = \begin{bmatrix} N_{1,x} & N_{2,x} & N_{3,x} \\ N_{1,y} & N_{2,y} & N_{3,y} \end{bmatrix}, \tag{23}$$

here $N_{i,j}$ represents the shape function associated with node $i$ of the element in the direction $j$. The vector $\boldsymbol{u}^e = [u_1, u_2, u_3]$ is comprised of the $u$ function values at the nodal locations of the triangle. Here we note that $\{x, y\}$ is actually a local coordinate system of the triangle that will depend on its orientation. In general, the gradient with respect to the global coordinates $\boldsymbol{X} = \{X, Y, Z\}$ can be obtained by changing the local to global coordinate systems with a rotation matrix $\boldsymbol{R}^e$: $(\nabla_{\boldsymbol{X}} u)^e = \boldsymbol{R}^e \boldsymbol{B}^e \boldsymbol{u}^e$. However, in all the applications of this manuscript this rotation matrix cancels. In the Eikonal equation, we can write the norm of the gradient as $\sqrt{\boldsymbol{u}^{eT} \boldsymbol{B}^{eT} \boldsymbol{R}^{eT} \boldsymbol{R}^e \boldsymbol{B}^e \boldsymbol{u}^e}$, but for any rotation matrix its inverse corresponds to its transpose, such that $\boldsymbol{R}^{eT} \boldsymbol{R}^e = \boldsymbol{I}$. From a physical perspective this makes sense: the norm of a vector is the same regardless the coordinate system. The rotation matrix involved in the operation $\nabla N_i \cdot \nabla N_j$ that defines the Laplacian also cancels following the same argument. We end this section by providing the specific form of the $\boldsymbol{B}^e$ matrix for a triangular element. To make the description more concise we introduce the following notation: $x_{ab} = x_a - x_b$ and $y_{ab} = y_a - y_b$, where $x_i$ and $y_i$ refer to the local coordinates of the node $i$ in the triangle. We first introduce the area of the triangle $A^e = (x_{13} y_{23} - x_{23} y_{13})/2$. Then, we can define

$$\boldsymbol{B}^e = \frac{1}{2A^e} \begin{bmatrix} y_{23} & y_{31} & y_{12} \\ x_{32} & x_{13} & x_{21} \end{bmatrix}. \tag{24}$$

## B    COMPARISON TO OTHER METHODS

Other than traditional PINNs, we compare to two other methods that are designed to work on complex geometries: a graph convolutional network Gao et al. (2022) and a physics-informed neural network that exactly imposes boundary conditions with signed distance functions Sukumar & Srivastava (2022).

### B.1    GRAPH CONVOLUTIONAL NEURAL NETWORK

The authors of this approach Gao et al. (2022) propose to use a graph convolutional network to approximate the solution of a partial differential equation. The input of this network will be locations of the nodes of the mesh. In general, convolution in a graph can be expressed as $g_{\boldsymbol{\theta}} \star \boldsymbol{x} = \boldsymbol{U} g_{\boldsymbol{\theta}} \boldsymbol{U}^T$, where $g_{\boldsymbol{\theta}}$ is a filter in Fourier domain and $\boldsymbol{U}$ are the eigenvectors of the graph Laplacian Welling & Kipf (2016). This operator is different from Laplace-Beltrami operator, which we use here, in the sense that it only encodes topological information about neighbouring nodes, but it completely lacks geometrical information. It is clear from the application of the graph convolution that it represents a linear combination of features. The result is then passed through a non-linear activation function and the convolution operation can be applied multiple times to create a deep graph-convolutional network. To avoid performing the eigen-decomposition of the graph Laplacian, it has been proposed to use Chebyshev polynomials Defferrard et al. (2016). The degree of the polynomial $K$ represents the

maximum separation of nodes that will be reached by the convolution. The authors set $K = 10$. Our approach is fundamentally different from the graph convolution, even though they are both related to the Laplacian. The graph convolution acts locally and propagates information to neighboring nodes Welling & Kipf (2016). Similarly, partial differential equations (PDEs) consider local differential operators, as certified by their sparsity when discretized with finite elements. However, the solution operator of such PDEs is global, and hence its finite dimensional counterpart is dense. Another way to see this is that local changes in the boundary conditions of the PDE yield global changes in the solution. For this reason, graph convolutional networks require either high order filters or multiple layers to propagate information across all nodes in the graph. In our approach, we encode global topological and geometric information about the domain in the Laplace-Beltrami eigenfunctions. These functions provide a suitable basis to approximate the solution of partial differential equations, which we non-linearly combine with a fully connected neural network.

In Gao et al. (2022), the authors also compute the operators used in the underlying partial differential equations using the finite element method, which makes it a perfect candidate for comparison against our method. We compare both methods on the coil example with the Eikonal equation, which is detailed in Section 3.1. To approximately match the number of parameters used in $\Delta$-PINN, we use 3 hidden layers with 16 filters each, with ReLU activations for the Chebyshev convolutions with $K = 10$. We train both methods for the same number of iterations and we verified that both methods reached stable value of the loss. It is worth mentioning that training the graph convolutional network takes significantly longer than $\Delta$-PINNs because it does not allow for a mini-batch implementation: the output at one location potentially depends on every input node of the graph. In the coil example, with only 1,546 nodes, $\Delta$-PINNs is 9x faster to train than the graph convolutional network. This speed-up would only grow in more complex meshes, as the size of the mini-batch can be fixed for $\Delta$-PINNs and will grow for the graph convolutional network.

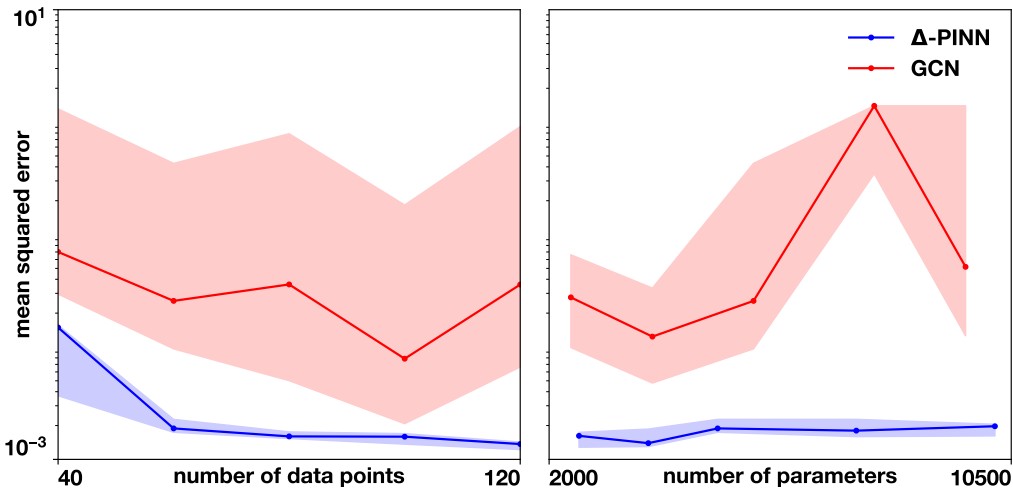

Figure 6: **Ablation study comparing to graph convolutional networks**. We assess the test in the coil example varying the amount of data points provided with a fixed architecture (left) and vary the amount of parameters of each methodology when training with 60 data points. We repeat each run 5 times with different random initialization of the parameters and show the minimum and maximum error with filled band and the median with a solid line.

We finalize this comparison with an ablation study, by comparing the performance of the two methods in the coil example. For all cases, we repeat the experiment with 5 different random initialization of the parameters. We compare the effect of adding more data points (Figure 6, left) for fixed architectures used in Section 3.1. We see that for all cases, $\Delta$-PINNs outperforms the graph convolutional network, and we see a decrease in error as we add more data points. This behavior is lacking for the graph convolutional network, which increases the error when 120 data points are used. We also assess the effect of the number of parameters in both methods when training with 60 data points (Figure 6, right). For $\Delta$-PINNs, we use only one hidden layer and 50 eigenfunctions, and vary the

amount of neurons in the hidden layer. For the graph convolutional networks, we use 3 hidden layers and vary the size of these layers. In $\Delta$-PINNs, we see very little influence in the error if we change the number of parameters. For the graph neural network, we see that is very sensitive to the architecture choice in a non-trivial way. We also see than in general, the graph convolutional network is much more sensitive to changes in initialization than $\Delta$-PINNs.

### B.2 EXACT IMPOSITION OF BOUNDARY CONDITIONS WITH SIGNED DISTANCE FUNCTIONS

In Sukumar & Srivastava (2022), the authors focused on solving forward problems with PINNs. Typically, the boundary conditions are enforced through a term in the loss function, here shown in equation 1. To avoid this approach, which may lead to errors, the authors propose the following structure to approximate the solution of the partial differential equation

$$\hat{u}(\boldsymbol{x}, \boldsymbol{\theta}) = NN(\boldsymbol{x}, \boldsymbol{\theta})\phi(\boldsymbol{x}) + g(\boldsymbol{x}), \tag{25}$$

where $NN(\cdot, \boldsymbol{\theta})$ represents a fully connected neural network with trainable parameters $\boldsymbol{\theta}$, $\phi(\cdot)$ is signed distance function, which is zero at the boundary, and $g(\cdot)$ is a function that interpolates the boundary conditions. With this construct, $\hat{u}$ satisfies the boundary conditions for any parameters $\boldsymbol{\theta}$. The authors propose efficient ways to estimate the signed distance function $phi(\cdot)$ and they propose to use transfinite interpolation for $g(\cdot)$, which we implemented. We test this method in the heat sink example, detailed in Section 3.2. For this method, we treat the data at the boundary as boundary conditions, which are then interpolated. In order to match the amount of parameters of $\Delta$-PINNs, we use 3 hidden layers with 100 neurons each and we train for the same number of iterations. In Figures 3 and 4, we see that this approach improves significantly the performance of traditional PINNs, however it is not able to match the performance of $\Delta$-PINNs.

We also note that the scope of this method is much narrower than $\Delta$-PINNs. So far it has only been demonstrated to work on 2D examples and there is no direct path to make it work in surfaces in 3D, which is one of the strengths of $\Delta$-PINNs. Also, it does not allow for noise in the boundary conditions, because the interpolation $g(\cdot)$ will exactly impose the noisy measurements.

## C  POISSON EQUATION WITH EXACT EIGENFUNCTIONS

In this example, we select a case that should favor traditional PINNs: solving a Poisson equation in a simple square domain. Here, we fabricate the following solution

$$u(x, y) = \left(x^2 - 1\right)\left(y^2 - 1\right)\exp\left(-(x - y)^2/l^2\right) \tag{26}$$

which can be seen in the left panel of Figure 7. Then, we define the following boundary value problem to be tackled by PINNs.

$$\begin{aligned} \Delta u(\boldsymbol{x}) &= f(\boldsymbol{x}), \, \boldsymbol{x} = \{x, y\} \in \mathcal{B} \\ u(\boldsymbol{x}) &= 0, \, \boldsymbol{x} \in d\mathcal{B}. \end{aligned} \tag{27}$$

The domain $\mathcal{B}$ is defined by a square in $[-1, 1] \times [-1, 1]$. The expression for $f(\boldsymbol{x})$ can be obtained by applying the Laplacian to the manufactured solution $u(\boldsymbol{x})$. In this domain, we can compute the eigenfunctions of the Laplace operator with homogeneous Neumann boundary conditions analytically, which correspond to

$$v_{l,m}(x, y) = \cos\left(\pi\frac{lx}{2}\right)\cos\left(\pi\frac{my}{2}\right), \tag{28}$$

with $l, m = 1, ..., \sqrt{N}$, where $N$ is the total number of eigenfunctions. We note that the boundary value problem described in equation 28 has Dirichlet boundary conditions, such that these eigenfunctions do not provide a suitable basis to solve this problem. In this case, we only provide data at the boundary $\boldsymbol{x} \in d\mathcal{B}$ and the values of $f(\boldsymbol{x})$ in the domain $\boldsymbol{x} \in \mathcal{B}$, effectively solving the forward boundary value problem shown in (28). The partial differential equation loss in this example corresponds to

$$MSE_{\text{PDE}}(\boldsymbol{\theta}) = \sum_i^R \left(\sum_{j \in N(i)} A_{ij}\hat{u}_j - f_i\right)^2, \tag{29}$$

where the index $i$ represents the nodes of the mesh inside the domain $\mathcal{B}$, $j \in N(i)$ represents the neighbor nodes of the node $i$, including itself. The term $\hat{u}_j$ represents the output of the neural network at the nodal locations $j$ that depends on trainable parameters $\boldsymbol{\theta}$, and $f_i$ corresponds to $f(\boldsymbol{x}_i)$, with $\boldsymbol{x}_i$ the position of node $i$. For traditional PINNs, we use automatic differentiation to compute $\Delta \hat{u}$ and construct the loss term $MSE_{\text{PDE}}$. We use neural networks with 3 hidden layers of 100 neurons and train with ADAM for 50,000 iterations.

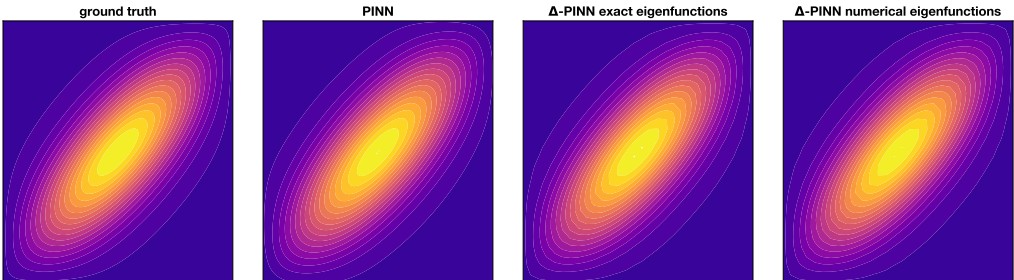

Figure 7: **Poisson example.** The ground truth manufactured solution, the approximation of traditional PINNs, $\Delta$-PINNs using the exact eigenfunctions as input and $\Delta$-PINNs using numerically approximated eigenfunctions by the finite element methods. For both cases of $\Delta$-PINNs, we use 100 eigenfunctions and an element size = 0.066 to discretize the operators.

We perform a sensitivity study to understand the influence of the number of eigenfunctions and the size of the discretization used to evaluate the operator $\Delta[\cdot]$ and the eigenfunctions computed numerically. We vary the number of eigenfuctions computed analytically and numerically from 9 to 400, and the size of finite elements between 0.02 to 0.2. Figures 7 and 8 summarize the results of this experiment. First, we note that traditional PINNs perform very well as expected, obtaining a mean squared error of $1.72 \cdot 10^{-5}$. Then, we observe that the results of $\Delta$-PINNs are sensitive to both the number of eigenfuctions and the discretization. When using the exact, analytical eigenfunctions, we are able to obtain low errors if we use more than 9 eigenfunctions for certain discretizations. When using the numerically obtained eigenfunctions, the same is true, expect for the case of 400 eigenfunctions. In this case, we cannot use a bigger discretization than 0.1 because the number of nodes becomes less than 400. The discretization size has a bigger impact on the errors for both cases. There is a sweet spot between 0.066 and 0.1 where most cases produce low error. The error when the discretization is big is expected, since the Laplace operator will not be accurately approximated. The error when the discretization is small is not straightforward to explain, but we hypotesize that in these cases the $MSE_{\text{PDE}}$ term becomes less important during training as the differences for $\hat{u}$ become smaller as the scale of the element decreases. In Figure 7, we can see qualitatively that both PINNs and $\Delta$-PINNs can successfully solve this boundary value problem.

## D  REGRESSION WITH $\Delta-$NN

In this section we test the capabilities of a neural network that takes as input the Laplace-Beltrami eigenfunctions to perform a regression task on a manifold. In this case, we do not include any physics that may act as a regularization of the predicted function. For this benchmark, we compare against Gaussian process regression with a Matérn kernel, which can be approximated on the manifold using the Laplace-Beltrami eigenfunctions as well Borovitskiy et al. (2020). We choose a dragon geometry 9,057 vertices and 18,110 triangles. We create a synthetic ground truth function by computing the geodesic distance from a vertex $d(\boldsymbol{x})$:

$$f(\boldsymbol{x}) = \sin\left(\pi \frac{d(\boldsymbol{x})}{d_{max}}\right), \tag{30}$$

where $d_{max}$ corresponds to the maximum geodesic distance from the selected vertex. We train with 50, 100, 500 and 1000 data points, randomly selecting them. We repeat this process five times and report the median error. For the neural network ($\Delta-$NN), we train using 9, 25, 49, 100, 225 or 400 eigenfunctions. We use one hidden layer of 100 neurons and hyperbolic tangent

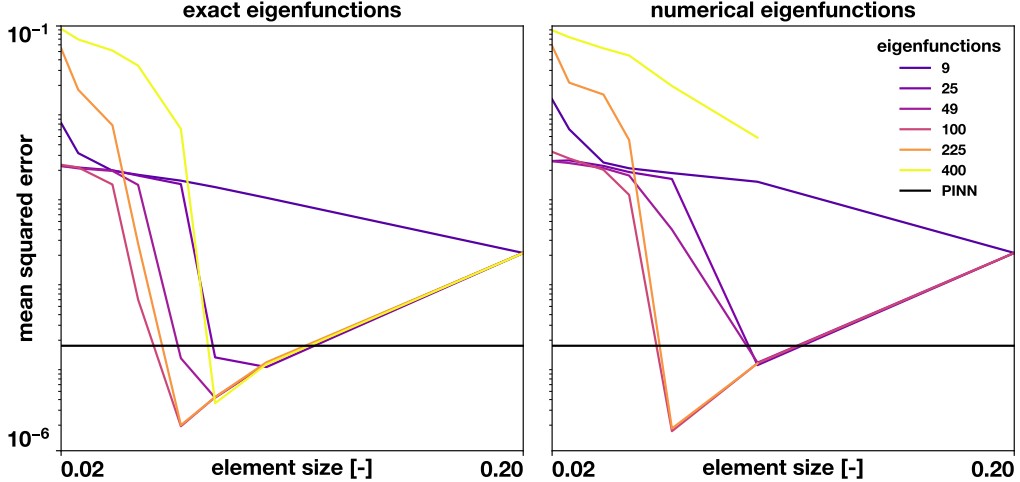

Figure 8: **Poisson example sensitivity analysis**. We run the Poisson example varying the two hyper parameters introduced by $\Delta$-PINNs: the number of eigenfunctions used as input and the element size used to discretize the operators. We test $\Delta$-PINNs using the exact eigenfunctions and the ones approximated by finite elements to understand the impact of this approximation.

activations. For the Gaussian process, we select $\nu = 3/2$ for the Matérn kernel and approximate it with 1000 eigenfunctions. The results are shown in Figure 9. In the lower right panel, we can see that the accuracy of $\Delta-$NN depends on the number of eigenfunctions used as input. When there are few data points available, it is preferable to have less eigenfunctions of lower frequencies to avoid overfitting. As the number of data points increases, more capacity is needed to approximate the data and the error decreases when more eigenfunctions are used. Nonetheless, using a high number of eigenfunctions, such as 225 and 400, never produces good results. On the other hand, Gaussian process regression excels in approximating this smooth function and always provides lower errors than $\Delta-$NNs. We also show in Figure 9, top row, a comparison of the output produced by $\Delta-$NN and Gaussian process when training with 100 points. In general, where there is data, both methods do a good job approximating the function. However, in the absence of data points, such as the horns of the dragon, $\Delta-$NN tends to predict more extreme values that increase the overall error. In this case, the overall mean squared error drops from $6.8 \cdot 10^{-2}$ $\Delta-$NN to $1.8 \cdot 10^{-2}$ for Gaussian process regression. We see that, in general, $\Delta-$NNs tend to overfit the data, and the addition of known physics across the domain has a regularization effect that improves the prediction, as seen in Figure 2 and 4.

# E $\Delta$-PINNS ON LARGE MESHES

In this section we show the time that it takes to compute the eigenfunctions of the Laplace-Beltrami operator of different sizes. We use the FEniCS library Logg et al. (2012) to assemble the operators, and the SLEPc library Hernandez et al. (2005) to solve the eigen-problem. We take 14 geometries from the open source repository https://github.com/alecjacobson/common-3d-test-models/ and compute 100 eigenfunctions. We use only 2 Intel(R) Xeon(R) CPUs (2.20GHz). The results are summarized in Figure 10. For the largest mesh with 134,345 nodes, the assembly of the operators and solving the eigen-problem took less than 35 seconds. This amount of nodes can be used to describe complex geometries. We also see excellent scaling with respect to the number of nodes. In this regard, training both traditional PINNs and $\Delta$-PINNs remains the largest computational burden in the process, and the computational cost of computing the eigenfunctions can be considered as a simple pre-processing step.

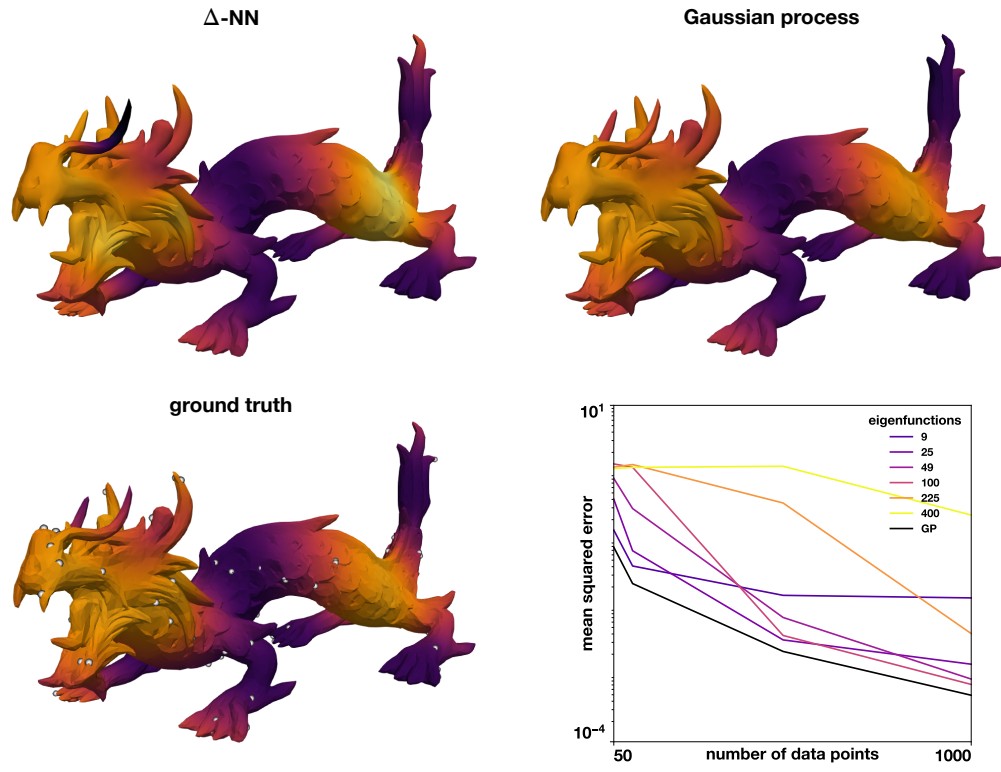

Figure 9: **Regression on a dragon.** We show the results when 100 data points for $\Delta-$NN using 25 eigenfunctions and Gaussian process. The ground truth is shown in the lower left, along with the training points, shown in white. The error comparison for different levels of data availability and different number of eigenfunctions for $\Delta-$NN are shown in the lower right panel.

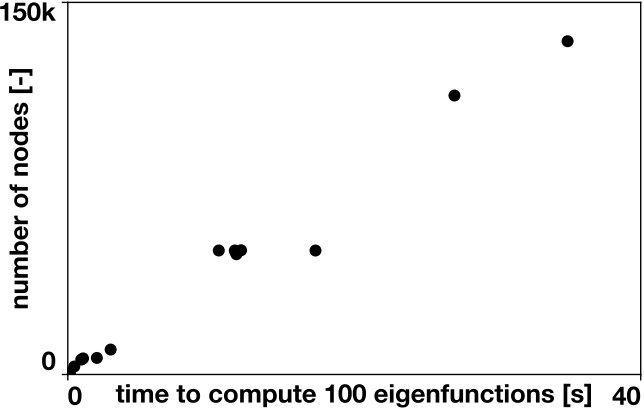

Figure 10: **Scaling of the eigenfunction computation**. We compute 100 eigenfunctions for a variety of meshes.

We finalize this section solving a partial differential equation on a large mesh with 99970 triangles and 49987 nodes from the model called Lucy. We set the following screened Poisson equation:

$$\Delta u - u \;=\; f \tag{31}$$
$$f \;=\; (\sin(3\pi x) + \cos(3\pi y) + \sin(3\pi z))/3, \tag{32}$$

where $\{x, y, z\}$ are the Cartesian coordinates of nodes. We note that this equation does not require boundary conditions to be solved. We generate the ground truth solution using finite elements, shown in Figure 11, left. The source term $f$ is shown in Figure 11, right. We train $\Delta$-PINN using 100 eigenfunctions, 2 hidden layers of 100 neurons. We train for 20,000 iterations with batch size 100. We reformulate the partial differential equation in an energy form for the $MSE_{\mathrm{PDE}}$. We obtain excellent agreement with the ground truth solution as shown in Figure 11, obtaining a mean squared error of $2.15 \cdot 10^{-4}$. Thus, we can see that $\Delta$-PINNs works well for large meshes, and that the eigenfunctions can be computed quickly.

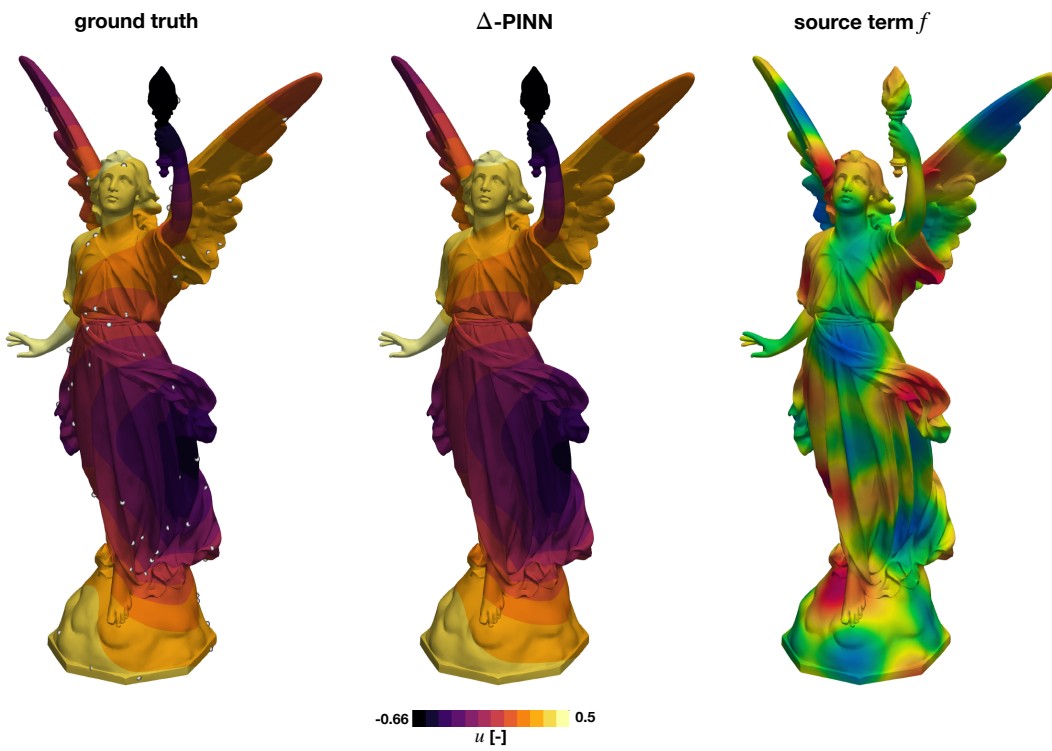

Figure 11: **Solving a partial differential equation on a large mesh**. We train $\Delta$-PINN with 100 data points on a mesh with 99970 triangles and 49987 nodes.

