# OpenReview forum: "$\Delta$-PINNs: physics-informed neural networks on complex geometries"
_ICLR.cc/2023/Conference — Submitted to ICLR 2023_

### Official Review · Reviewer_yokT · 2022-10-21

**Confidence:** 3
**Correctness:** 4
**Technical Novelty And Significance:** 3
**Empirical Novelty And Significance:** 4
**Recommendation:** 8

**Clarity, Quality, Novelty And Reproducibility:**

As mentioned in the strengths section, the clarity and quality of writing is good but have room for improvement. The method is not necessarily wholly novel, but it is a culmination of prior work that has clearly taken this field of research to the next level. It is clear that the authors have taken steps to ensure their work is reproducible by making mentions of accompanying code and referencing the prior work on which the code is built.

**Strength And Weaknesses:**

The paper is overall strong; the language is clear, the problem being solved is made obvious, and the effectiveness of the proposed solution is presented in a convincing manner. Though I am not an expert on the mathematics being utilized, I was still able to follow the logical procession of the equations even if I could not grasp the details. The figures and visualizations were not confusing, and the captions served well in clarifying what little doubt the figures may have presented.

However, the one potential weakness I see is the mention of the finite element method. I happen to know what that is, and I am sure an audience with a general physics background would know what that is, but given that this is a CS-focused paper, perhaps a paragraph explaining what the finite element method is and how it works would be helpful to offer further understanding for those readers that would not know what it is. Given that understanding the finite element method is important for understanding the methodology of the paper, I believe this would be a worthwhile addition.

In addition, a concrete example of how physics informed network is used in Section 2.1 will help the readers to understand the contribution better. I happen to work in this domain, so I know the meaning of the three MSEs. However, I doubt if people outside of our domain understand the need of using these three terms.


**Summary Of The Paper:**

This paper introduces ∆-PINNs, a novel form of physics-informed neural networks (PINNs) that leverages the eigenfunctions of the Laplace-Beltrami operator of the manifold to represent the geometry of a 3D solid. This advancement would expand the use of PINNs from simple geometric domains to more complex forms. The paper goes into great detail on how this method is implemented and presents multiple test cases comparing against baselines such as previous PINNs or ground-truth, showcasing that the proposed method outperforms prior PINNs and more closely matches ground-truth observations.

**Summary Of The Review:**

This paper makes a good contribution towards physics-informed neural net, by leveraging the eigenfunctions of the Laplace-Beltrami operator of the manifold to represent the geometry of a 3D solid. The paper possesses good clarity of expression, a well-explained proposal, and does not stumble in convincing the reader and expanding knowledge. Other than a small suggestion to include an explanation of the finite element method and relevant applicational domain, I believe this paper is worthy of being accepted.

---

> ### Author Response · Authors · 2022-11-18
> **Revision**
>
> We thank the reviewer for their comments and appreciate the positive feedback. We have included a section in Appendix A explaining the finite element method in the context of the proposed methodology. We agree that adding an example of how physics-informed neural networks are used will be helpful, but considering that we have a strict 9 page limit, we prefer to not include it to preserve other parts of the text that we believe are more important. We also think that through the numerical examples the reader will familiarize themselves with the uses of physics-informed neural networks.

---

### Official Review · Reviewer_vdFL · 2022-10-24

**Confidence:** 3
**Correctness:** 3
**Technical Novelty And Significance:** 3
**Empirical Novelty And Significance:** 3
**Recommendation:** 5

**Clarity, Quality, Novelty And Reproducibility:**

The paper is clear and well written, the description is precise and easy understanding.

There are some minor problems in the paper: in Figure 3 the result of “\delta-PINNs not informed by the heat equation” is not included, the caption should be amended as well as the following explanation text; the review of the Laplace-Beltrami eigenfunctions in Section 4 Discussion is suggested to put in Section 1 Introduction.

The novelty is fair. It is the first time the Laplace-Beltrami eigenfunctions is used in PINNs, while the methodology itself is not novel and has been applied in other neural networks.

The reproducibility is good.

**Strength And Weaknesses:**

Strength:
The proposed methodology enables the PINNs to solve problems defined on complex geometric domains with good accuracy.

Weakness:
1. The differential operators in the PDEs cannot be computed directly from PINNs using AD after applying the positional encoding method. Instead, they are computed numerically on the linear finite elements.
2. Computing the eigenfunctions of the Laplace-Beltrami operator on large meshes can be very expensive.
3. The performance of the proposed method strongly depends on the number of eigenfunctions used for positional encoding, while this quantity is a hyper-parameter needs to be tuned case by case.

**Summary Of The Paper:**

This paper proposed a novel positional encoding mechanism to inform PINNs about the topology of the domain, expanding the effectiveness of PINNs to complex geometric domains. The proposed positional encoding mechanism represents the coordinates of the input geometry with the eigenfunctions of the Laplace-Beltrami operator of a manifold, creating an input space for PINNs that represents the input geometry. Extensive numerical experiments (mainly in 2D) were carried out to compare the proposed methodology against traditional PINNs, showing excellent agreement with ground truth for cases where traditional PINNs fail.

**Summary Of The Review:**

The paper proposed a novel positional encoding mechanism to inform PINNs about the topology of the domain, expanding the effectiveness of PINNs to complex geometric domains. While the weakness of the proposed methodology is as distinguishable as its strength, the computation cost and the implementation complexity of the proposed methodology can be higher than the traditional PINNs, making it less promising.

---

> ### Author Response · Authors · 2022-11-18
> **Revision**
>
> We thank the reviewer for the constructive comments. We have significantly revised the paper in order to address the potential weaknesses mentioned by the reviewer. We added comparison to other geometry-infomed neural networks, such as graph convolutional network and a fully connected network with signed distance functions, explained in detail in Appendix B. Regarding the specific weaknesses mentioned by the reviewer, we would like to add:
> 1. Although it is true that we cannot use automatic differentiation directly on the network, there are multiple finite element libraries that support automatic differentiation and have been integrated with machine learning libraries, see here for example: https://github.com/barkm/torch-fenics. In our case, we coded the operators ourselves to keep the code self-contained and because for the triangular elements that we use they are only simple matrices, which we detail in Appendix A. In the future, we plan to interface our code with a finite element library to gain the ability to quickly obtain the operators for any partial differential equation and any finite element discretization. We also note that the competing method of graph convolutional network also relies on finite element differentiation. And, while PINNs with signed distance functions can use automatic differentiation, this method cannot be applied to surfaces in 3D.
> 2. Regarding the cost of computing the eigenfunctions, we see that if we use specialized eigensolvers this becomes a minor pre-processing step. We have added Appendix E, where we compute 100 eigenfunctions for meshes of different sizes and we see that for a mesh with 135,000 nodes, which is more than enough to represent a complex geometry, it only takes 35 seconds using modest hardware. For most applications, we do not see the computation of the eigenfunctions as a limitation of the method. We have also added in Appendix E an example on a large mesh with a complex geometry to show that the method scales well in larger problems.
> Regarding the selection of an adequate number of eigenfunctions, we would like to say that finding the appropriate capacity is a universal problem for all neural networks. Even traditional PINNs will fail with the wrong architecture. In particular, for all positional encodings some additional parameters need to be tuned. However, this technique allows neural networks to learn functions that are impossible to approximate with a standard architecture. In the case of Cartesian positional encodings, such as Fourier features, they allow fully-connected neural networks to represent images, signed distance functions, radiance fields, etc.. With the positional encoding that we propose, neural networks can learn functions on surfaces in 3D, which we use to represent the solution of partial differential equations.
> 3. Finally, we would like to add that the number of eigenfunctions needed will depend on the physics of the problem and the complexity of the geometry. For instance, using 50 eigenfunctions works well for coil and bunny examples, which involve the Eikonal equation. In this way, we can transfer the hyperparameters to different cases. We also show in Appendix B, that the competing method, a graph convolutional network, has a much larger dependency on the architecture than our method.
>
> We would also like to comment that the computational cost of $\Delta$-PINNs is only marginally higher than traditional PINNs. As we have said, the cost of computing the eigenfunctions is very small and the training time for the coil example, for instance, is less than a minute using CPU. Both PINNs and $\Delta$-PINNs allow mini-batch implementations, which allows for good scaling as the problems get larger. The competing method of graph convolutional networks does not allow for mini-batch (see Appendix B), which slows the training process. In the coil example, training the graph convolutional network took 9 times longer than $\Delta$-PINNs with a modest size of ~1500 nodes. Finally, we would like to add that even though $\Delta$-PINNs come with additional complexities, they are worth it because they allow us to solve problems that are out of reach for traditional PINNs.
>
> We also thank the reviewer for the comments regarding the presentation. We have now fixed the caption of Figure 3.

---

### Official Review · Reviewer_ebMA · 2022-10-25

**Confidence:** 4
**Correctness:** 2
**Technical Novelty And Significance:** 2
**Empirical Novelty And Significance:** 1
**Recommendation:** 3

**Clarity, Quality, Novelty And Reproducibility:**

Some aspects of the experimentation are not clear.
-In Figure 1, which/how many eigenfunctions did you use to get a solution of the Laplace-Beltrami operator on the manifold. Also the figure caption needs to be more descriptive. (e.g. the color bar for geodesic is confusing to have before explaining in section 3 that the solution to the Eikonal equation can be interpreted as a geodesic distance to any point in manifold)
-How did you select the 50 eigenfunctions for input into the NN? Perhaps there is a principled way to learn which and how many eigenfunctions could be included in the mesh.
-IN section 2.3 the authors need to elaborate on how they computing the numerical approximation and extracting the gradients – this seems to be a key part of the method
-- Bottom of page 5:  these numbers should be listed in a table or with the figure.





**Strength And Weaknesses:**

Strengths:
The authors’ clearly motivate the need to incorporate geometric/topological information into their method. A nice feature of encoding the NN with eigenfunctions of the Laplace-Beltrami Operator over the domain is being able to hand select the eigenfunctions based on the magnitude of the eigenvalue given prior knowledge about the nature of the problem

Weaknesses:
However, they don't justify their choice of computing eigenfunctions of a continuous Laplace-Beltrami  is their approach. As they say, there is no closed form available, and they could have gone with a graph approach with a graph laplacian. In this setting there are neural networks that directly compute the eigenvectors of graph laplacians, for instance PowerEmbedd from this paper https://arxiv.org/abs/2209.12054.  Thus this whole approach could have been done with a GCN. This is not included as a comparison. In general, numerical experiments aren’t extremely convincing – need more ablations/comparisons to other types of networks but PINN and non-PINN ones such as GNNs, transformers, etc.

**Summary Of The Paper:**

In this paper, the authors propose a PINN-style network where the physical quantities are computed on a manifold. The manifold geometry of the data is represented via a positional encoding based on the eigenfunctions of the Laplace-Beltrami operator of the manifold,  propose to represent the coordinates of the input geometry with a positional encoding. The Laplace-Beltrami operator as well as the eigenfunctions are estimated using a finite element solver.



**Summary Of The Review:**

At a high level the manuscript is well-principled and the authors explain conceptually how they incorporate geometry into the PINN. The paper does have drawbacks – one of the major concerns is the lack of comparisons to the other neural networks or geometric-informed PINNs that the authors mentioned. Additionally, the authors could provide more detail on how they numerically approximate the Laplace-Beltrami operator and the finite element scheme/package that was used. Overall, the idea is nice but the impact is not distinguished and its unclear how their method compares to others.

---

> ### Author Response · Authors · 2022-11-18
> **Revision**
>
> We thank the reviewer for their comments. We have significantly revised the manuscript and added comparison to a physics-informed graph convolutional network (GCN) and a physics-informed neural network informed by the geometry with signed distance functions. Our method outperforms the GCN, which is based on the graph Laplacian. We think that the main cause for this improvement is that graph convolution acts locally and propagates information to neighboring nodes. However, we know that solutions of partial differential equations are global, where the information given by the boundary conditions will affect the entire domain. For this reason, graph convolutional networks require either a high order filter or multiple layers to reach all the nodes. In our approach, we encode global topological and geometric information about the domain in the Laplace-Beltrami eigenfunctions. These functions provide a suitable basis to approximate the solution of partial differential equations, which we non-linearly combine with a fully connected neural network. Also, the geometrical information in the GCN is given by the input features, which are the coordinates of the mesh and topological information is given by the graph Laplacian. In our case, this geometrical information is already embedded in the Laplace-Beltrami operator. We have included a section in Appendix B to compare our methodology to other networks and performed additional ablation studies.
>
> Thanks for pointing us to the powerembed method, we missed it since it was published just 5 days before this conference’s deadline. The powerembed method agrees with our approach in the sense that the eigenfunctions of the graph operator encode global information, rather than the local message passing that is achieved by graph convolutions. Since we work with geometrical objects rather than generic graphs, we use the Laplace-Beltrami operator which contains both topological and geometrical information. The powerembed method approximately computes eigenfunctions during the forward pass. In our case, we can pre-compute the eigenfunctions with very little cost (less than 40 seconds for 135k nodes, see Appendix E), and then we can save time during training.
>
> Regarding the comment of a principled way to select the number of eigenfunctions, the reviewer raises a very interesting point. If we had the full field data (i.e. data on every node of the mesh) we could first perform a linear fit with different number of eigenfunctions and stop at the point of diminishing returns regarding the error. This would be similar to performing the Fourier transform on the manifold and dropping low energy components. However, having full field data is the exception in the context of physics-informed neural networks. In general, the partial differential equation being considered and the complexity of the geometry will influence the number of eigenfunctions needed and there is no simple way to predict it. In this sense, hyperparameter tuning seems to be the best option, although the number of eigenfunctions should remain useful for given physics and given mesh complexity. For instance, using 50 eigenfunctions work well for coil and bunny examples, which involve the Eikonal equation. Finally, we note that finding the appropriate capacity of a neural network is a universal problem, and traditional PINNs can also fail with the wrong architecture.
>
> We thank the reviewer for all the comments regarding the presentation. We have improved Figure 1 and its caption, we have included a section explaining the finite element method in Appendix A.

---

### Author Response · Authors · 2022-11-18
**Revision from the authors**

We would like to thank the reviewers for their feedback, as we believe the manuscript has improved substantially. Here we highlight the major changes to the manuscript:
- We have included a detailed explanation of the finite element method in the context of the proposed methodology in Appendix A.
- We have added comparisons to two other methods, a physics-informed graph neural network and a fully connected physics-informed neural network with signed distance functions. We performed additional ablation studies comparing to the graph convolutional network, showing better performance in different data regimes and network configurations. All the details are shown in Appendix B.
- We show that our method scales well for large meshes. We demonstrate that computing the eigenfunctions of the Laplace-Beltrami operator can be quickly done when using specialized libraries, taking in the order of seconds for large meshes. We also show an example of our method in a large mesh, showing excellent agreement with the ground truth solution. All the details are included in Appendix E.

---

### Decision · Program_Chairs · 2023-01-20

**Decision:**

Reject

**Justification For Why Not Higher Score:**

Some of these points of criticism raised by the reviewers question the general approach and methodology proposed, and these issues could not be convincingly addressed in the rebuttal/discussions.

**Justification For Why Not Lower Score:**

N/A

**Metareview: Summary, Strengths And Weaknesses:**

For this paper, we had two negative and one positive review. Summing up all reviews, comments an discussions, I came to the conclusion to vote for rejection. The reasons are:
-  It seems that the general idea based on the eigenfunctions of a continuous Laplace-Beltrami operator is not motivated in a clear and transparent way, and even after the rebuttal, it remained somewhat unclear, why one could not simply use a graph-based approach.
- in general, it seems that there are many other alternative approaches to solving the problem considered, and the comparison experiments provided were not fully convincing in this regard.
- there might be pronounced performance issues due to the possible need for including many eigenfunctions. Although the authors commented on this issue in a detailed way, I am still not fully convinced that their approach will be computationally advantageous over alternative ones (which -- from a conceptual viewpoint -- also might be well-motivated and applicable) .

**Summary Of Ac-Reviewer Meeting:**

After a conversation with reviewer yokT, I had the impression that the over-all positive perception of this work was mostly caused by the novelty of the general idea, and not so much by specific formal concepts.  In particular, I could not find additional arguments that would address the above mentioned issues in a convincing way.